# Acclimatization of a coral-dinoflagellate mutualism at a $CO_2$ vent

Fiorella Prada[1,2,13,14], Silvia Franzellitti [2,3,14], Erik Caroselli [1,2✉], Itay Cohen[4], Mauro Marini [2,5], Alessandra Campanelli[5], Lorenzo Sana[3], Arianna Mancuso [1,2], Chiara Marchini[1,2], Alessia Puglisi[3], Marco Candela [2,6], Tali Mass [7], Franco Tassi[8,9], Todd C. LaJeunesse [10✉], Zvy Dubinsky[11], Giuseppe Falini [2,12] & Stefano Goffredo [1,2✉]

Ocean acidification caused by shifts in ocean carbonate chemistry resulting from increased atmospheric $CO_2$ concentrations is threatening many calcifying organisms, including corals. Here we assessed autotrophy vs heterotrophy shifts in the Mediterranean zooxanthellate scleractinian coral *Balanophyllia europaea* acclimatized to low pH/high $pCO_2$ conditions at a $CO_2$ vent off Panarea Island (Italy). Dinoflagellate endosymbiont densities were higher at lowest pH Sites where changes in the distribution of distinct haplotypes of a host-specific symbiont species, *Philozoon balanophyllum,* were observed. An increase in symbiont C/N ratios was observed at low pH, likely as a result of increased C fixation by higher symbiont cell densities. $\delta^{13}C$ values of the symbionts and host tissue reached similar values at the lowest pH Site, suggesting an increased influence of autotrophy with increasing acidification. Host tissue $\delta^{15}N$ values of 0‰ strongly suggest that diazotroph $N_2$ fixation is occurring within the coral tissue/mucus at the low pH Sites, likely explaining the decrease in host tissue C/N ratios with acidification. Overall, our findings show an acclimatization of this coral-dinoflagellate mutualism through trophic adjustment and symbiont haplotype differences with increasing acidification, highlighting that some corals are capable of acclimatizing to ocean acidification predicted under end-of-century scenarios.

[1] Marine Science Group, Department of Biological, Geological and Environmental Sciences, University of Bologna, Via F. Selmi 3, 40126 Bologna, Italy. [2] Fano Marine Center, The Inter-Institute Center for Research on Marine Biodiversity, Resources and Biotechnologies, Viale Adriatico 1/N, 61032 Fano, Italy. [3] Animal and Environmental Physiology Laboratory, Department of Biological, Geological and Environmental Sciences, University of Bologna, via S. Alberto 163, 48123 Ravenna, Italy. [4] The Interuniversity Institute for Marine Sciences in Eilat, PO Box 469 Eilat 88103, Israel. [5] Institute of Biological Resources and Marine Biotechnology, National Research Council (CNR), Largo Fiera della Pesca 2, 60125 Ancona, Italy. [6] Unit of Microbiome Science and Biotechnology, Department of Pharmacy and Biotechnology, University of Bologna, 40126 Bologna, Italy. [7] Department of Marine Biology, The Leon H. Charney School of Marine Sciences, University of Haifa, Haifa, Israel. [8] Department of Earth Sciences, University of Florence, via la Pira 4, Firenze, Italy. [9] Institute of Geosciences and Earth Resources (IGG), National Research Council of Italy (CNR), via la Pira 4, Firenze, Italy. [10] Department of Biology, The Pennsylvania State University, 208 Mueller Laboratory, University Park, PA 16802, USA. [11] The Mina and Everard Goodman Faculty of Life Sciences, Bar-Ilan University, Ramat-Gan 52900, Israel. [12] Department of Chemistry "Giacomo Ciamician", University of Bologna, 40126 Bologna, Italy. [13] Present address: Environmental Biophysics and Molecular Ecology Program, Department of Marine and Coastal Sciences, Rutgers University, New Brunswick, NJ 08901, USA. [14] These authors contributed equally: Fiorella Prada, Silvia Franzellitti. ✉email: erik.caroselli@unibo.it; tcl3@psu.edu; s.goffredo@unibo.it

The defining feature of the Anthropocene era[1] is the emergence of human activities as a driving force of global change[2], which is occurring at a rate that raises concerns whether organismal adaptation can keep pace with rapidly changing environmental conditions[3]. Stressors associated with ocean warming and acidification are among the most direct and pervasive anthropogenic changes for marine biota, including corals[4]. The decrease in pH from ca. 8.2 prior to the industrial revolution to ca. 8.1 with a doubling of $CO_2$ is leading to a gradual decline in the saturation state of calcium carbonate in seawater[5]. This phenomenon has been projected to negatively impact the ability of corals to calcify[6,7]. However, empirical evidence, in terms of e.g., natural selection of tolerant bacterial and/or dinoflagellate symbionts, differential regulation of environmental stress response genes[8,9], suggests an underappreciated capacity of corals to acclimate and genetically adapt to environmental change[10]. Indeed, these ancient organisms have survived, evolved, and adapted throughout hundreds of millions of years of global climate change[11–14].

The symbiosis between scleractinian corals and their dinoflagellate microalgae (family Symbiodiniaceae), commonly referred to as zooxanthellae, has been extensively studied[15–17]. Zooxanthellae significantly contribute to the energy budget of the host by providing photosynthetically fixed carbon[16] while recycling host respiration and excretion by-products[18]. In such symbiosis, both carbon and nitrogen can be obtained via heterotrophy and autotrophy and are recycled between the host and dinoflagellate symbionts[19,20]. Generally, the symbiosis provides most of the carbon needed for respiration[21], while predation on zooplankton and particulate organic matter are still needed to meet nitrogen and phosphorus requirements[16]. Nonetheless, the relative contribution of heterotrophy versus autotrophy on host nutrition varies across species, populations, environments, and/or with ontogenesis[22].

A critical limitation for many experimental studies has been replicating the rate (decades) and biological scales (ecosystems) at which ocean acidification operates. Natural $CO_2$ vents acidify the surrounding seawater creating carbonate chemistry conditions that mimic future ocean acidification predictions[23–25], even if with wide short-term variability[26–28]. By investigating natural populations living along transects radiating from the sources of $CO_2$, these systems allow substituting time for space, providing valuable insights on acclimatization and adaptation to ocean acidification[29]. This study was conducted on natural populations of the zooxanthellate scleractinian coral *Balanophyllia europaea* living at a $CO_2$ volcanic vent near Panarea Island (Italy). This underwater crater at 10 m depth releases persistent gaseous emissions (98–99% $CO_2$ without instrumentally detectable toxic compounds), resulting in a stable pH gradient at ambient temperature[30] with ocean acidification conditions projected for 2100 under conservative and worst-case IPCC scenarios[31,32]. With decreasing pH, *B. europaea* shows a decline in population density[33] and net calcification rates, the latter as a result of increased skeletal porosity, whereas linear extension rate is preserved[24], allowing the coral to reach size at sexual maturity[28]. Moreover, *B. europaea* maintains unchanged skeletal calcium carbonate polymorph, organic matrix content, aragonite fiber thickness and skeletal hardness, calcifying fluid pH, and gross calcification with decreasing pH[34].

The aim of this study was to further investigate the physiological acclimatization of this coral-dinoflagellate mutualism with increasing acidification, by assessing the relative contribution of photosynthetically *versus* heterotrophically derived nutrition (autotrophic/heterotrophic ratio). This was compared with other key physiological parameters (i.e., photosynthetic efficiency, symbiont cell densities, and chlorophyll concentration) and with

the distribution of genetically distinct haplotypes of the dinoflagellate symbiont *Philozoon balanophyllum* along the gradient. We expected to find an overall physiological acclimatization of the symbiont, allowing the coral to sustain energetically costly processes (e.g., maintaining elevated calcifying fluid pH, reproduction) that in previous studies were found to be constant along the natural pH gradient[24,28,34].

## Results

**Environmental parameters.** Among measured seawater parameters (pH, total alkalinity, temperature, and salinity), only pH changed across Sites (Kruskal–Wallis test, $H = 38$, $df = 2$, $p = 0.000$; Supplementary Table 1). Even though fluctuations were observed for pH, the average value decreased from 7.97 (95% CI: 7.95–7.99) at Site 1, to 7.86 (95% CI: 7.83–7.90) at Site 2, to 7.64 (95% CI: 7.55–7.77) at Site 3 (Mann-Whitney U test, $p = 0.000$; Fig. 1; Supplementary Table 1). Average aragonite saturation ($\Omega_{arag}$) decreased by almost 30% from Site 1 (average: 3.2, 95%CI: 3.1–3.4) to Site 3 (average=2.3; 95% CI: 2.2–2.4) (Mann–Whitney U test, $p = 0.000$; Fig. 1; Supplementary Table 1). No significant difference in $\Omega_{arag}$ was observed between Sites 2 and 3 (Mann–Whitney U test, $p = 0.432$; Supplementary Table 1).

Phosphate and inorganic nitrogen (Nitrate+Nitrite) were homogeneous across Sites (Kruskal–Wallis test, $H = 1.192$, $df = 2$, $p = 0.551$ and $H = 3.082$, $df = 2$, $p = 0.214$, respectively), while sulphate was significantly lower at Site 3 compared to Sites 1 and 2 (Mann–Whitney U test, $p = 0.009$ and $p = 0.016$, respectively; Supplementary Table 2). No significant difference in sulphate was observed between Sites 1 and 2 (Mann-Whitney U test, $p = 0.293$; Supplementary Table 2).

**PAM fluorometry measurements.** Photosynthetic yield of the dinoflagellate symbionts, measured on 7–25 haphazardly chosen corals per Site (Supplementary Table 3), did not change along the gradient in any of the assessed time intervals. However, yield was

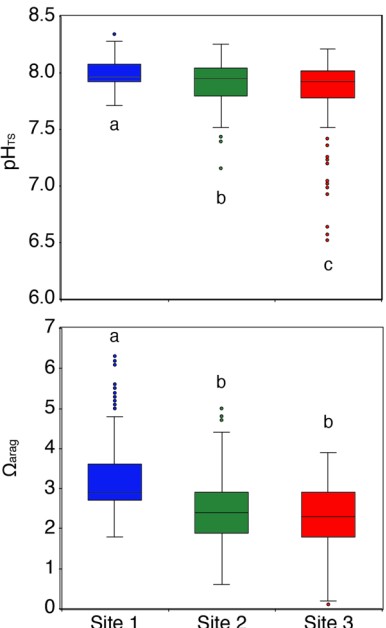

**Fig. 1 Ranges of measured pH_TS and Ω_arag values showing consistent decreases from Site 1 to Site 3.** The boxes indicate the 25th and 75th percentiles and the line within the boxes mark the medians. Whisker length is equal to 1.5 × interquartile range (IQR). Circles represent outliers. Different letters indicate statistical differences ($p < 0.05$; number of observations: pH_TS = 185-198 per Site; Ω_arag = 184–195 per Site).

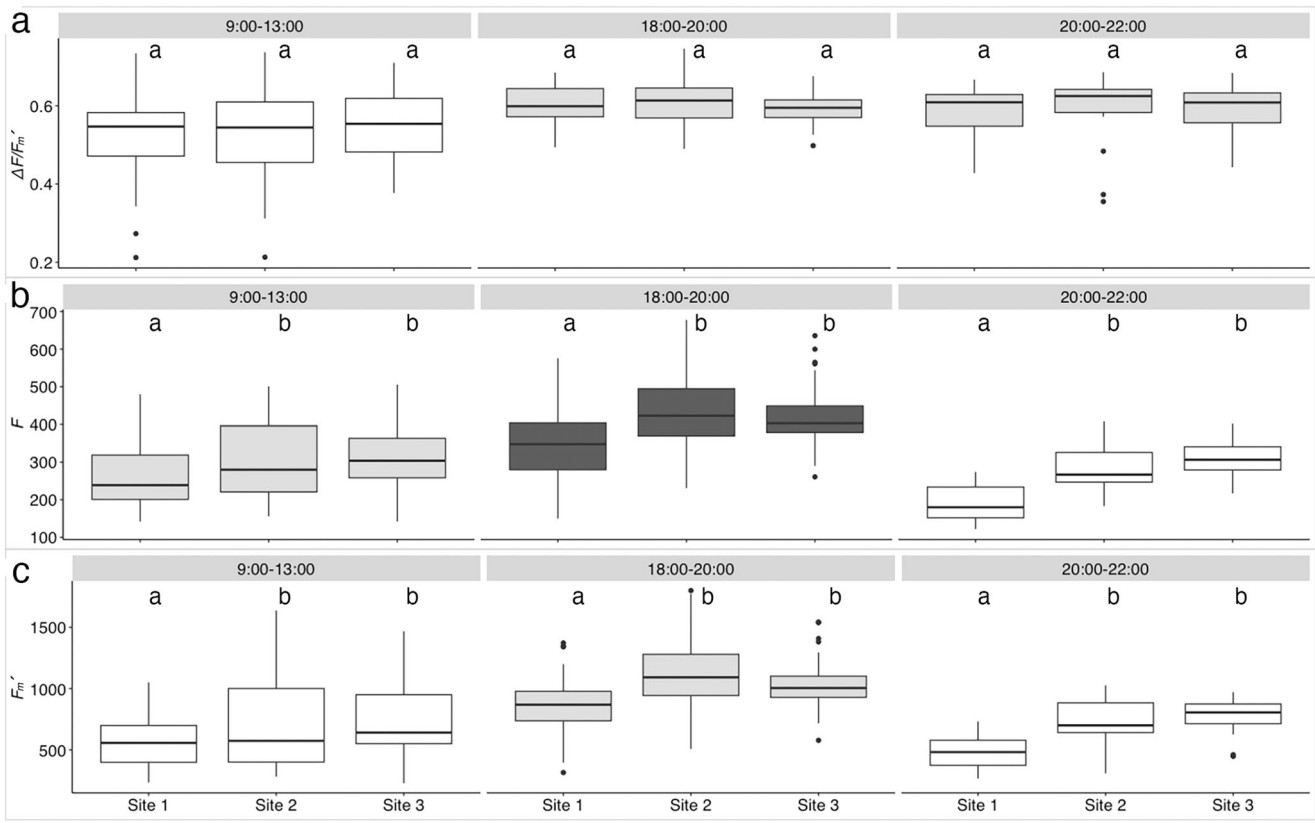

**Fig. 2 Boxplots of fluorescence parameters obtained by PAM measurements at the three Sites and at three time intervals. a** Effective quantum yield ($\Delta F/F_m'$), **b** minimum fluorescence ($F$), and **c** maximum fluorescence ($F_m'$). For each parameter, significant differences ($p < 0.01$) between time intervals are indicated by different colors (significantly higher mean value is ranked as white->light gray->dark grey). The boxes indicate the 25th and 75th percentiles and the line within the boxes mark the medians. Whisker length is equal to 1.5 × interquartile range (IQR). Circles represent outliers. Different letters indicate statistical differences ($p < 0.05$; number of specimens measured is reported in Supplementary Table 3).

significantly higher during the low light time intervals 18:00–20:00 and 20:00–22:00 than in 9:00-13:00 (Fig. 2; Supplementary Tables 4 and 5). Minimum ($F$) and maximum fluorescence ($Fm'$) were significantly higher at Sites 2 and 3 compared to Site 1 at all time intervals. Minimum fluorescence ($F$) increased from 20:00–22:00 to 9:00–13:00 to 18:00–20:00, while maximum fluorescence ($Fm'$) was significantly higher in the time interval 18:00–20:00 than in the other time intervals (Fig. 2; Supplementary Tables 4 and 5).

**Variations in symbiont cell density and chlorophyll-a concentration.** Increasing numbers of symbiont cells from Site 1 to Site 3 were observed along the gastroderm (endoderm) of the mesenteries in histological sections (Fig. 3a–f). Indeed, the average number of symbiont cells per area was higher at Sites 2 and 3 compared to Site 1 (LSD post hoc test: $p = 0.002$; Fig. 3g). No significant difference in the number of symbiont cells was observed between Sites 2 and 3 (LSD post hoc test: $p = 0.968$; Fig. 3g). Chl-a concentration per symbiont cell was homogenous among studied Sites (One-way ANOVA, $F_{2,9} = 0.359$, $p = 0.708$; Fig. 3h). The average chl-a concentration, expressed as chl-a amount per polyp area (pg/mm$^2$), was higher at Sites 2 and 3 compared to Site 1 (LSD post hoc test: $p = 0.015$ and $p = 0.053$, respectively; Fig. 3i). No significant difference in chl-a amount per polyp area was observed between Sites 2 and 3 (LSD post hoc test: $p = 0.449$; Fig. 3i).

**$\delta^{13}C$ and $\delta^{15}N$ variability and C/N ratios.** The $\delta^{13}C$ values for both dinoflagellate symbionts and host tissue did not show

significant statistical differences along the gradient (symbionts: Kruskal-Wallis test, $H = 2.346$, d$f = 2$, $p = 0.309$; tissue: One-way ANOVA, $F_{2,9} = 1.243$, $p = 0.334$; Fig. 4; Supplementary Table 6). However, at Site 3 (pH=7.64, pCO$_2$ = 1161) $\delta^{13}C_h$ and $\delta^{13}C_z$ reached similar values in some specimens (Fig. 4; Supplementary Fig. 1) and the proportion of carbon in the host supplied by the symbionts[35] reached the highest values (Fig. 4).

$\delta^{15}N$ values in symbionts were homogeneous among Sites (One-way ANOVA, $F_{2,9} = 1.001$, $p = 0.405$; Fig. 4; Supplementary Table 6), whereas $\delta^{15}N$ in coral tissue showed heavier values in Site 1 compared to Sites 2 and 3 (LSD post hoc test: $p = 0.000$). No significant difference in host tissue $\delta^{15}N$ was observed between Sites 2 and 3 (LSD post hoc test: $p = 0.181$).

Host tissue C/N ratio was significantly lower in Sites 2 and 3 compared to Site 1 (LSD post hoc test, $p = 0.000$; Fig. 4; Supplementary Table 6). No significant difference in host tissue C/N ratio was observed between Sites 2 and 3 (LSD post hoc test: $p = 0.237$). Symbiont C/N ratios increased from Site 1 to Sites 2 and 3 (LSD post hoc test, Site 1 vs Site 2: $p = 0.005$, Site 1 vs Site 3: $p = 0.000$, Site 2 vs Site 3, $p = 0.002$; Fig. 4; Supplementary Table 6).

**PsbA$^{ncr}$ haplotypes of *Philozoon balanophyllum*.** Analysis of the *psbA$^{ncr}$* identified *Philozoon balanophyllum* in each sample of *B. europaea* analyzed ($N = 4$ per Site) (Supplementary Dataset 1). Moreover, each individual *B. europaea* harboured a single *P. balanophyllum* haplotype. When compared phylogenetically with sequences obtained from *P. balanophyllum* from other regions in the Tyrrhenian and Adriatic Seas, *psbA$^{ncr}$* haplotypes

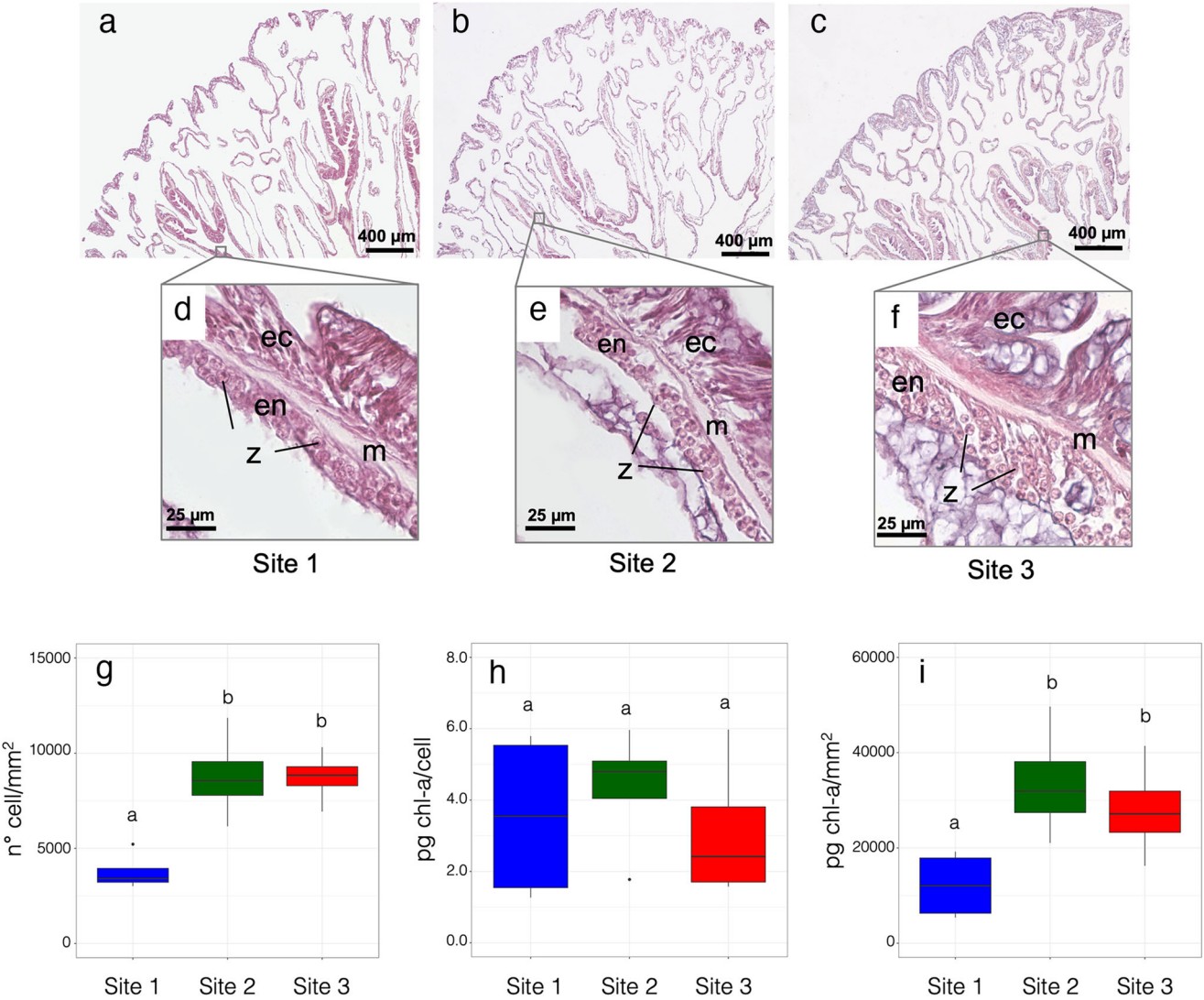

**Fig. 3 Symbiont cells in _B. europaea_ polyps sampled along the Panarea pH gradient. a–c** Histological transverse sections of representative polyps in Sites 1-3 (x 4 magnification). **d–f** Symbiont cells within the endoderm of the mesentery highlighted by x40 magnification. z: zooxanthellae; ec: ectoderm; m: mesoglea; en: endoderm. **g** Symbiont cell density and **h** chlorophyll-a (chl-a) concentration normalized over cell count or **i** polyp area. The boxes indicate the 25th and 75th percentiles and the line within the boxes mark the medians. Whisker length is equal to 1.5 × interquartile range (IQR). Circles represent outliers. Different letters indicate statistical differences ($p < 0.05$; number of corals = 4 per Site).

from the acidified Sites 2 and 3 grouped to a distinct genetic lineage statistically distinct from all other haplotypes representative of the species (Fig. 5: Supplementary Dataset 1). Haplotype sequences from the less acidified environment of Site 1 and from samples obtained around the Tyrrhenian and Adriatic Sea were more similar to each other (Fig. 5).

## Discussion

As calcifying organisms, corals are threatened by increasing absorbance of atmospheric $CO_2$ into Earth's oceans. Our research indicates that populations of _B. europaea_ adjust to decreasing pH through changes in symbiont cell densities and their haplotype identities, which appear to affect the amount of photosynthetically fixed carbon relative to heterotrophically acquired carbon in the host. Therefore, certain coral-dinoflagellate mutualisms have the capacity to acclimatize to ocean acidification by various ways.

_Balanophyllia europaea_'s relative dependence on autotrophy and heterotrophy for nutrient acquisition appears in part

contingent on ocean pH. The $\delta^{13}C$ values in host tissues and dinoflagellate symbionts are indicative of greater photosynthetic inputs at the low pH/high $pCO_2$ Sites[36,37]. The higher seawater $pCO_2$ values at Site 3 probably account for lighter $\delta^{13}C$ values in the symbionts[22]. The latter result is consistent with previous findings reporting a greater net autotrophic input to the carbon budget under low pH/high $pCO_2$ conditions in the temperate sea anemone _Anemonia viridis_ investigated at a similar $CO_2$ vent system at Vulcano Island (Italy)[37]. Moreover, greater autotrophy through increased symbiont abundances, enhanced C fixation, and less heterotrophically derived nitrogen may also explain the increase in carbon relative to nitrogen concentration in dinoflagellate symbiont cells from Sites 2 and 3 compared to Site 1[38]. Symbiont cell densities and host tissue thickness are known to vary in corals photo-acclimating to seasonal changes in solar irradiance[39–41]. The increase in host tissue (endoderm) thickness from ~15 μm in Site 1 (pH 8.0) to ~40 μm in Site 3 (pH 7.6) and the corresponding increase in symbiont cell densities is similar to winter phenotypes observed in tropical corals[40]. Tissue

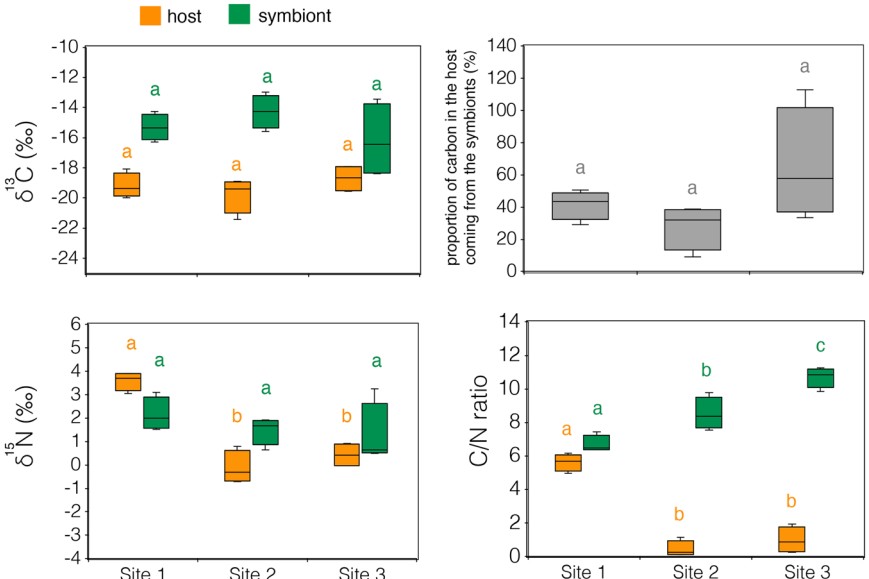

**Fig. 4 $\delta^{13}$C, $\delta^{15}$N, C/N ratios, and proportion of carbon in the host coming from the symbionts in *B. europaea* tissue from Sites 1 (control), 2 (intermediate pH) and 3 (low pH).** The proportion of carbon in the host supplied by the symbionts was calculated using the formula: $\delta^{13}C_{tissue} = (\delta^{13}C_{symbiont})x + (1-x)(\delta^{13}C_{zooplankton/POC})$. We assume $\delta^{13}C_{zooplankton/POC} = -22$‰[37]. The boxes indicate the 25th and 75th percentiles and the line within the boxes mark the medians. Whisker length is equal to 1.5 × interquartile range (IQR). Different letters indicate statistical differences ($p < 0.05$) between Sites (number of corals = 4 per Site).

thickening has been observed in other coral species experiencing low seawater pH conditions[42,43] and may therefore constitute a general phenotypic response to ocean acidification. Moreover, enhanced translocation of photosynthetically fixed carbon to the host as a result of high symbiont densities could contribute to tissue growth and increased energy reserves. Indeed, previous findings have shown that energy derived from photosynthesis may have up to 20-fold greater effect on tissue production compared to skeleton deposition[44].

Zooxanthellate corals typically associate with dinoflagellate species that raise the functional tolerance of the mutualism when existing in thermally stressful environments[45,46]. Physiological constraints related to the pH gradient across the study transect may explain the non-random distribution of distinct haplotypes of *Philozoon balanophyllum* (formerly temperate Clade A) in *B. europaea*. The small but fixed differences in *psbA*ncr between populations over a distance of 30 meters or less is greater than the genetic differences in this symbiont distributed over many hundreds of kilometers across the Mediterranean basin (Fig. 5). It is compelling to assume the related haplotypes at Sites 2 and 3 may have attributes (allelic combinations) better fit to the lower pH conditions, possibly contributing to the increase in symbiont cell density observed at the low pH Sites compared to Site 1. However, this possibility will require additional physiological characterization[47].

Tissue C/N ratios in the low pH/high pCO$_2$ Sites were unusually low compared to ratios from the control Site (Fig. 4). The latter were consistent with values typically found in marine organisms (from 6.5 to 8.7)[48]. C/N ratios can reach values around 2 in marine particulate organic matter[49,50] and even drop below 1 in sediments characterized by significant amounts of fixed nitrogen[51]. Surprisingly, we found that the C/N ratio of the symbionts followed an opposite trend compared to the tissue, increasing with decreasing pH. Indeed, assuming that the elemental composition of an animal reflects that of its diet[52], this observation may, at first, appear contradictory: If the algae feed the host, and the symbiont's C/N ratio rises, should not the host's

C/N ratio follow suit? However, this logic negates the potential for the host to receive nutritional input from multiple sources. Many corals harbor bacteria capable of fixing N$_2$ gas (diazotrophs)[53] and a growing body of literature indicates that ocean acidification may promote the enrichment of dinitrogen fixing bacterial communities in natural microbiomes[8,54,55]. Indeed, recent work at our study site found that the prevalence of genes associated with N$_2$ fixation, as well as the production of N storage molecules, increases in the *B. europaea* microbiome under low pH conditions[56]. Increased assimilation of diazotroph-derived N by the coral is supported by our isotopic data (Fig. 4). At low pH Sites, the $\delta^{15}$N of host tissue was near 0‰. This value is significantly lower than typically reported for zooxanthellate coral species[57] and strongly suggests that diazotroph N$_2$ fixation is occurring within the coral tissue/mucus[58,59]. Transcriptomic and proteomic studies aimed at identifying expressed proteins involved in nitrogen fixation (e.g., nitrogenase)[60] as well as isotope tracer experiments (e.g., $^{15}$N-labeled dinitrogen) aimed at quantifying N$_2$ fixation[61] are needed to have a better understanding of this process in *B. europaea* along the Panarea pH gradient.

Figure 6 summarizes the main skeletal and physiological changes displayed by *B. europaea* acclimatized to low pH/high pCO$_2$ conditions at the Panarea CO$_2$ vent. While exposure to low pH creates more porous skeletons, linear extension rates are maintained[24], allowing corals to reach size at sexual maturity and reproduce normally[28], despite possessing a more fragile skeleton[24]. The acquisition of symbiont strains potentially better adapted to acidified conditions could further contribute to increased nutrient cycling and animal growth. Bacteria involved in nutrient-cycling and N$_2$ fixation may play a crucial role in supplementing coral-Symbiodiniaceae symbiosis with additional nitrogen under ocean acidification, helping sustain the higher photosynthetic rates expected under acidified conditions[62–64]. Likewise, it has been hypothesized that photosynthetically fixed carbon supplied by the dinoflagellates may serve as an energy source for N$_2$ fixation in cyanobacterial

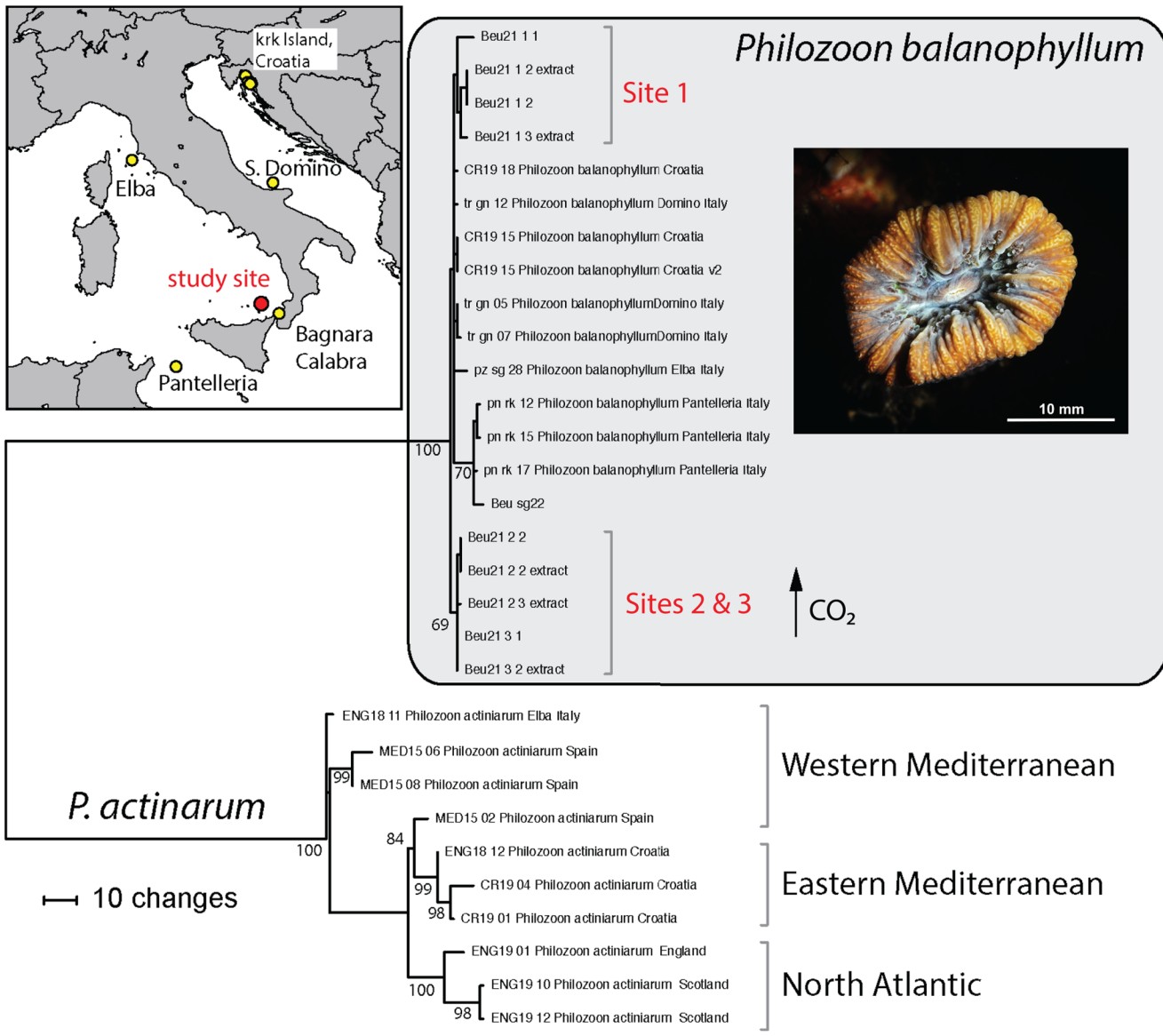

**Fig. 5 *Philozoon balanophyllum* phylogeny.** *Philozoon balanophyllum* halpotypes in *B. europaea* specimens from Sites 1-3 along the Panarea pH gradient and from around the Tyrrhenian and Adriatic Seas. Haplotypes of *Philozoon actiniarum*, the symbiont of the sea anemone *Anemonia viridis*, common to the Mediterranean, are provided as the outgroup. (Photograph by Francesco Sesso).

symbionts in corals[60], suggesting that in zooxanthellate corals, photosynthesis and $N_2$ fixation may be, in some cases, inter-dependent. Moreover, enhanced $N_2$ fixation by diazotrophs living in association with the coral tissue/mucus[56] may partially explain the observed increase in dinoflagellate symbiont cell densities in *B. europaea* at the low pH Sites[65], and viceversa. Enhanced translocation of photosynthetically fixed carbon as a result of high dinoflagellate symbiont cell densities may help sustain the high costs of diazotroph $N_2$ fixation under increased acidification. Indeed, dinoflagellates supply glycerol as an energy source for cyanobacterial symbionts found in *Montipora cavernosa*, providing a steady supply of reductant (e.g., NADPH) and ATP for dinitrogen fixation in the cyanobacteria[60]. Additionally, $N_2$ fixation by coral-associated diazotrophs, coral calcification, reproduction, and tissue thickening are energy-intensive processes that probably compete for energy deriving from photosynthesis within the coral holobiont[66]. Thus, massive $N_2$ fixation, together with main-tenance of coral reproduction and thickening of coral tissue

occurring at the low pH Sites could be absorbing most of the photosynthetic energy from the symbiotic algae, creating an energy deficit that could possibly explain the previously observed decline in net calcification rates of this species along the same pH gradient[24,34]. Taken together, current and previous findings highlight the importance of the interactions among all the components of the holobiont to unveil how and to what degree corals will endure ocean acidification predicted for the end of the century.

## Methods

**Environmental parameters**. Temperature, salinity, and pH (NBS scale) were measured at three Sites respectively 34 m (Site 1), 13 m (Site 2) and 9 m (Site 3) away from the center of the crater with a multi-parametric probe (600 R, YSI Incorporated) powered from a small boat and operated by SCUBA divers. Total alkalinity was determined by Gran titration (888 Titrando) from bottom-water samples collected at the three Sites and poisoned by adding 1% of saturated $HgCl_2$ shortly after collection[33]. Certified reference materials (Batch 187) provided by Andrew Dickson (Scripps Institution of Oceanography, La Jolla, CA) were used to ascertain the quality of results obtained. Environmental data were collected during several expeditions between 2010–2013[24,32,33] and 2019-2020 (this study).

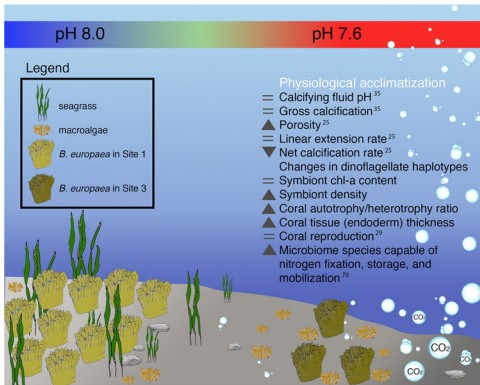

**Fig. 6 Conceptual scheme summarizing the effects of life-long physiological acclimatization to low pH/high pCO2 conditions in _B. europaea_ at the Panarea CO2 vent.** Under low pH conditions, coral population density decreases[33], net calcification is depressed, while linear extension rate is maintained constant, allowing the coral to reach critical size at sexual maturity and reproduce. Moreover, an overall rearrangement of the coral holobiont is documented by: (i) an increase in symbiont cell density, triggered by thickening of the coral tissue and the establishment of novel dinoflagellate haplotypes possibly better adapted to lower pH conditions, and (ii) an increase, within the coral tissue/mucus in microbial communities capable of dinitrogen fixation as well as N storage and mobilization. Variations displayed by corals living at average pH 7.6 compared to corals at average pH 8.0 are shown with grey symbols and are listed in the following order: skeleton (from micro to macro), symbionts (from micro to macro), trophic strategy, reproduction, tissue/mucus microbial community. Darker and lighter shades of green in the coral sketches represent higher and lower symbiont cell density. The color scale bar highlighting the seawater pH change across the gradient does not match the color scale of pH test strips. Image was assembled using Adobe Photoshop CC (19.1.6).

Measured pH$_{NBS}$ were converted to the total scale using CO2SYS software[67]. Mean pH$_{TS}$ (back-transformed hydrogen ion concentrations) was calculated for each Site and used with total alkalinity, salinity and temperature to calculate carbonate chemistry parameters using the software CO2SYS with referenced dissociation constants[68–70]. Bottom water samples for dissolved inorganic nutrients were collected at the four Sites using 100 ml plastic bottles (two replicates for each Site) and frozen at −20 °C. Inorganic nitrogen (nitrite-NO$_2$ + nitrate-NO$_3$) and phosphate (orthophosphate-PO$_4$) were determined using a colorimetric method[71,72]. Absorbances were measured with an AxFlow quAAtro AutoAnalyzers (N-detection limit/sensitivity: 0.006/0.001 μM; P-detection limit/sensitivity: 0.006/0.001 μM). Sulphate was measured by ion chromatography using a Metrohm 761 compact IC.

**PAM fluorometry measurements.** Photosynthetic yield was estimated by measuring PSII (photosystem II) fluorescence using a Diving PAM (Pulse Amplitude Modulator) (Heinz Walz GmbH). Measurements were taken in July 2019 and February 2020 on a total of 353 B. europaea specimens naturally living at Sites 1-3 at the Bottaro CO$_2$ vents off Panarea Island[24,33]. During each dive, 7–25 corals per Site (Supplementary Table 3) were haphazardly selected for photosynthetic yield measurement. Divers used very dim, red-filtered diving flashlights to avoid biases during the measurements in the time interval 20:00-22:00 (i.e. in the dark). During the day, measurements were performed between 9:00–13:00 and 18:00–20:00. For each coral, the probe of the Diving PAM was pointed vertically to the coral mouth at a distance of 1 cm and the PSII fluorescence measurement was taken after holding the sensor steady until fluorescence values were steady. The Diving PAM produces weak red flashes (measuring light) and detects the initial core fluorescence returns at 780–800 nm ($F$), upon measurement a strong white light saturates PSII reaction centers thus most of the measuring light is dissipated via fluorescence (maximal fluorescence, $Fm'$). From these parameters, maximal (night time) and effective quantum yield (day time) ($\Delta F/Fm'$) were calculated using the following equation[73]:

$$\Delta F/Fm' = (Fm' - F)/Fm'$$

**Separation of dinoflagellate symbionts from the coral host tissue.** Coral samples ($N = 8$ corals per Site) were collected in July 2019 using a hammer and a chisel by SCUBA diving and placed in label plastic containers. All samples were transported in ice to the laboratory and frozen at −20 °C until further processing. Coral tissue was removed with an airbrush connected to a reservoir of phosphate buffer saline (PBS) solution filtered through a 0.22 μm filter, and the skeleton was kept for further analysis. Separation of symbiont cells and coral tissue was performed by mechanical homogenization and centrifugation of the homogenate (5000 x g for 5 min at 4 °C) using a protocol adapted from previous studies, that accounts for the validity of the separation method from a qualitative (no tissue/debris contamination in the symbiont pellet) and quantitative (complete symbiont cell precipitation) point of view[36,74–76]. Following separation, aliquots of symbiont cell suspensions were processed for cell count and chlorophyll concentration assays. Tissue and symbiont suspensions were frozen at −20 °C, then lyophilized separately and stored at −20 °C prior to δ$^{13}$C and δ$^{15}$N stable isotope analyses.

**Symbiont cell density and chlorophyll-a concentration.** Symbiont cell density in the homogenate was determined by fluorescent microscopic counts (Nikon Eclipse Ti, Japan) using a hemocytometer (BOECO, Germany) and 5 replicate (1 mm$^2$ each) cell counts per sample ($N = 4$ corals per Site). Each replica was photographed both in brightfield and in fluorescent light using 440 nm emission to identify chlorophyll. Cell counting was performed using NIS-Elements Advanced Research (version 4.50.00, Nikon, Japan) with 0.5< Circularity <1, and the typical diameter parameter was to set between 5 and 15 μm. Chlorophyll-a (chl-a) concentration was measured in 2 ml of resuspended symbiont cell homogenate that was filtered onto a Whatman GF/C filter and incubated overnight with 1 ml 90% acetone at 4 °C. After incubation, the filter was manually homogenized, and the solution was filtered through a 0.22 μm syringe filter. A NanoDrop (Thermo-Fisher, United States) was used for spectrophotometric measurements at 630, 647, 664 and 697 nm wavelengths and absorbance values were used to calculate the chl-a concentration based on the following equation[77]:

$$chl - a[\mu g\ ml^{-1}] = -0.3319 * ABS630 - 1.7485 * ABS647 + 11.9442 * ABS664 - 1.4306 * ABS697$$

**Histological analyses and image acquisition.** Polyps collected for histology were immediately fixed in a formalin solution (10% formaldehyde in 37% seawater saturated with calcium carbonate) before being transferred to the lab. Samples were then postfixed in Bouin solution (composed of 15 ml picric acid saturated aqueous solution, 5 ml formaldehyde 37%, and 1 ml glacial acetic acid). After decalcification in EDTA and dehydration in increasing concentration of ethanol (from 80% to 100%), polyps were embedded in paraffin. Sections were cut at 7 μm intervals along the oral-aboral axis. Tissues were stained with Mayer's hematoxylin (Carlo Erba) and eosin (Sigma-Aldrich®)[78].

Histological observations of the coral tissue were performed using a light microscope NIKON Eclipse 80i coupled with a Nikon NIS-Elements D high-resolution digital camera. Sections of the three polyps were photographed at the same distance from the oral pole at 4x magnification. Subsequently, a detail of the mesentery was photographed at 40x magnification for the observation of the symbiont cells within the endoderm.

**δ$^{13}$C and δ$^{15}$N stable isotope analysis.** Analyses were performed at the Godwin laboratory for Paleoclimate Research, Dept. of Earth Sciences, Cambridge University (UK). Tissue and symbiont samples ($N = 4$ corals per Site) were analysed for percentage carbon and nitrogen, $^{12}$C/$^{13}$C and $^{14}$N/$^{15}$N using a Costech Elemental Analyser attached to a Thermo DELTA V mass spectrometer in continuous flow mode. Reference standards from IAEA in Vienna were analysed along with the samples. The dried sample/standard was carefully weighed into a tin capsule, sealed and loaded the auto-sampler. Reference standards were run at intervals throughout the sequence and these values were used to calibrate to the international standards for $^{14}$N/$^{15}$N (δ$^{15}$N air) and $^{12}$C/$^{13}$C (δ$^{13}$C VPDB). Precision of analyses is +/−0.05 % for C and N, better than 0.1 % for $^{12}$C/$^{13}$C and better than 0.1 % for $^{14}$N/$^{15}$N.

**Analyses of the resident dinoflagellate symbionts.** Frozen polyps of B. europaea ($N = 4$ corals per Site) were powdered with a mortar pestle using liquid nitrogen[79]. DNA was isolated with the Wizard Genomic DNA Purification kit (Promega) according to the manufacturer's instructions. Quality and quantity of extracted DNA was double checked using electrophoresis (0.8% agarose gel) and spectrophotometric measurements (λ=260 nm/280 nm).

The high resolution _psb_A non-coding region (_psb_A$^{ncr}$) from the chloroplast mini-circle genes of dinoflagellates[80,81] was amplified and then directly sequenced. The 'universal' primers psbAFor_1 (5′GCA GCT CAT GGT TAT TTT GGT AGA C 3′) and psbARev_1 (5′AAT TCC CAT TCT CTA CCC ATC C 3′), designed to amplify the psbA$^{ncr}$ for most Symbiodiniaceae[81], were used with the following PCR conditions: 94 °C for 2 min; then 40 cycles of 94 °C 10 s, 55 °C for 30 s and 72 °C for 2 min; and a final extension at 72 °C for 10 min. The internal primers Philozoon-psbAF (5′ATT TGG TTC ACA GCG CTT GG 3′) and Philozoon-psbAR (5′CCA TTT GAC TCC CAC ACT GGA) were also used for nucleotide sequencing through the middle region of the amplified fragment[82]. Direct Sanger sequencing on PCR amplified DNA was performed using Big Dye 3.1 reagents (Life Sciences) and the Applied Biosystems 3730XL instrument.

**Statistics and reproducibility**. Data were checked for normality using a Kolmogorov–Smirnov test ($N > 50$) and Shapiro-Wilk test ($N < 50$) and for homogeneity using Levene's Test. One-way analysis of variance (ANOVA) and the non-parametric Kruskal–Wallis equality-of-populations rank were used to assess differences in environmental parameters (number of observations for each environmental parameter are reported in Supplementary Table 1), symbiont cell density, chlorophyll-a concentration, $\delta^{13}C$, $\delta^{15}N$, C/N ratios, and amount of carbon translocated from the symbiont to the host among Sites ($N = 4$ corals per Site for all listed biological parameters). Where significant, pairwise comparisons between species were performed via LSD or Mann–Whitney U test. Data analyses were performed using SPSS Statistics 26.0 and GraphPad Prism 9 software. Due to the heteroskedastic dataset, mean minimum fluorescence ($F$), maximum fluorescence ($Fm'$) and effective quantum yield ($\Delta F/Fm'$) were compared among Sites and time intervals with a permutation multivariate analysis of variance (PERMANOVA)[83] based on Euclidean distances, using a crossed design with two fixed factors (factor "Site" with 3 levels: Site 1, Site 2, Site 3; factor "time interval" with 3 levels: 9:00–13:00, 18:00–20:00, 20:00–22:00) and 999 permutations (number of corals analyzed are reported in Supplementary Table 3). PERMANOVA analyses were performed with software Primer 6 (Primer-e Ltd).

Base calling on chromatograms was visually inspected for accuracy (Geneious v. 11.0.3) and the edited sequences aligned initially using the online application of ClustalW2 (http://www.ebi.ac.uk/Tools/msa/clustalw2/) ($N = 4$ corals per Site). Further adjustments to alignments were made upon visual inspection of the output file. Final edited sequences were deposited in GenBank. Phylogenetic analyses using Maximum Parsimony, confirmed with Maximum Likelihood, were conducted using the software PAUP (v. 4.0a136; Swofford, 2014) on aligned sequences. One-thousand bootstrap replicates were used to assess statistical significance of internal branching.

**Reporting summary**. Further information on research design is available in the Nature Portfolio Reporting Summary linked to this article.

## Data availability

All source data underlying the graphs presented in the main figures are reported in Supplementary Data 1 and 2. All data and materials produced by this study are available from the corresponding author upon request.

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

## Acknowledgements

The research leading to these results has received funding from the European Research Council under the European Union's Seventh Framework Programme (FP7/2007–2013)/ ERC grant agreement n° 249930—CoralWarm: Corals and global warming: the Mediterranean versus the Red Sea. Research project implemented under the National Recovery and Resilience Plan (NRRP), Mission 4 Component 2 Investment 1.4 - Call for tender No. 3138 of 16 December 2021, rectified by Decree n.3175 of 18 December 2021 of Italian Ministry of University and Research funded by the European Union – Next-GenerationEU. Project code CN_00000033, Concession Decree No. 1034 of 17 June 2022 adopted by the Italian Ministry of University and Research, CUP J33C22001190001, Project title "National Biodiversity Future Center - NBFC". LaJeunesse was supported by funding from the USA National Science Foundation (grant OCE-1636022). Bartolo Basile, Francesco Sesso and Eolo Sub diving center assisted in the field. The Scientific Diving School collaborated with the underwater activities. We are grateful to Francesco Sesso for the image of B. europaea. We are also thankful to Prof. Costantino Vetriani and Dr. Corday Selden from Rutgers University whose comments were instrumental in developing certain parts of the Discussion.

## Author contributions

S.G., G.F., and Z.D. conceived and designed the research. F.P., E.C., A.M., and S.G. collected the samples and performed the diving field work. F.P., S.F., E.C., M.M., A.C., L.S., C.M., A.P., T.M., F.T., T.C.L. performed the laboratory analyses. F.P., E.C., S.F., I.C., M.C., T.C.L. and S.G. analysed the data. F.P., S.F., and E.C. wrote the first draft. All authors contributed to writing the manuscript and participated in the scientific discussion.

## Competing interests

The authors declare no competing interests.
