## [Peer Review File · Communications Biology]

Reviewers' comments:

Reviewer #1 (Remarks to the Author):

Ocean acidification caused by a shift in ocean carbonate chemistry resulting from increased atmospheric CO₂ concentrations is a major threat for many calcifying organisms, including corals. Although there are numerous literatures regarding the acclimation of corals to low pH conditions, most of the studies are not on-site investigations. In current study, the authors examined the natural variability of carbon and nitrogen isotopes and physiological parameters in the zooxanthellate scleractinian coral *Balanophyllia europaea* along a natural pH gradient at Panarea Island, Italy (Mediterranean Sea). It provides valuable points on the acclimatization of this coral-algal mutualism through trophic adjustment and genotypic differences in the algal symbiont in seawater with different pH. It highlights that some corals are well capable of acclimatizing to the ocean acidification. Overall, the manuscript is well written that might absorb a broad interest of readers. I only have a minor comment. Figure 6 seems confusing and not quite relevant to the main topic of the manuscript. More explanation or deletion might be helpful for readers.

Reviewer #2 (Remarks to the Author):

Review of Prada et al 13271_O_rd9twc

Acclimatization of a coral-algal mutualism at a CO₂ vent

This study reports of the results of a field study looking at the zooxanthellate Scleractinian coral *Balanophyllia europaea* living along a natural CO₂ gradient. The objective was to determine the ability of this species to acclimatize trophically to reduced pH conditions by assessing the relative contribution of photosynthetically versus heterotrophically derived nutrition (autotrophic/heterotrophic ratio). This was compared with other key physiological parameters (i.e., photosynthetic efficiency, algal symbiont density, and chlorophyll concentration) and with the haplotype composition of the algal symbiont *Philozoon balanophyllum* along the gradient.

Lines 159-160 Please mention the basis for this separation method. Are the symbiont cells more or less dense than the coral host cells? Please cite the paper that establishes centrifugation as a reliable method for separating symbiont and host cells. I have seen this method of separation used before but never a publication to support the basis for a quantitative separation of host and symbiont cells.

Significant results include an increase in symbiont density between Site 1 (control) and Sites 2 and 3 (reduced pH). Chl a per unit area showed a similar pattern. These results suggest that production of photosynthetic assimilates would be greater in the corals found at Sites 2 and 3. The PAM data show no significant difference in photosynthetic efficiency of the symbionts across the gradient. This implies that the symbiont cells aren't acclimatizing to the reduced pH. Rather the coral host may be regulating the density of symbiont cells in such a way as to optimize the availability of photosynthetic assimilates under reduced pH conditions when energy expenditures for calcification and acid/base homeostasis will be higher. It is interesting that $\delta^{13}C$ data also support more translocation of photosynthetic assimilate in the Site 2 and 3 corals, although it is not clear to me that the difference (60% v 40%) is statistically significant. Are the authors able to test the statistical significance of this difference?

The massive reduction in host C:N ratio doesn't make much sense to me. If the C:N of the symbionts is increasing and there is an increase in the amount of that photosynthetic assimilate translocated to the host wouldn't you expect the C:N of the host to also increase (assuming that you are what you eat)? Also the extremely low C:N ratios of the host at Sites 2 and 3 seem questionable to me. Is it really possible for the bulk organic matter of coral host to have C:N composition of 1:1?

My conclusion is that this is a very interesting manuscript that should be published but there are some things that I have called attention to that should be resolved first.

Point-by-point response to Reviewer's comments

Reviewer #1

COMMENT

Ocean acidification caused by a shift in ocean carbonate chemistry resulting from increased atmospheric CO₂ concentrations is a major threat for many calcifying organisms, including corals. Although there are numerous of literatures regarding the acclimation of corals to low pH conditions, most of the studies are not on-site investigations. In current study, the authors examined the natural variability of carbon and nitrogen isotopes and physiological parameters in the zooxanthellate scleractinian coral *Balanophyllia europaea* along a natural pH gradient at Panarea Island, Italy (Mediterranean Sea). It provides valuable points on the acclimatization of this coral-algal mutualism through trophic adjustment and genotypic differences in the algal symbiont in seawater with different pH. It highlights that some corals are well capable of acclimatizing to the ocean acidification. Overall, the manuscript is well written that might absorb a broad interest of readers. I only have a minor comment. Figure 6 seems confusing and not quite relevant to the main topic of the manuscript. More explanation or deletion might be helpful for readers.

REPLY:

Agreed. We thank the reviewer for pointing this out. In light of this suggestion, we gave Figure 6 a lot of thought because we think it should truly summarize several years of work conducted on this species in this natural ocean acidification laboratory. We redesigned the entire figure to show at a glance the main findings from this study and how they complement previous research on *B. europaea* overall acclimatization strategy to low pH/high pCO₂ at the Panarea CO₂ seep. In light of these changes, we also significantly expanded the conclusions of the manuscript.

Reviewer #2

COMMENT

Lines 159-160 Please mention the basis for this separation method. Are the symbiont cells more or less dense than the coral host cells? Please cite the paper that establishes centrifugation as a reliable method for separating symbiont and host cells. I have seen this method of separation used before but never a publication to support the basis for a quantitative separation of host and symbiont cells.

REPLY:

Agreed. We now provide a more detailed explanation of the different steps used to separate the symbionts from the tissue, specifying that the denser symbionts are pelleted while the host tissue is the supernatant. We also cite the relevant literature establishing centrifugation as a reliable method for separating symbiont and host cells (see below). Changes were made in Lines 230-238.

Xu, S. et al. Spatial variations in the trophic status of *Favia palauensis* corals in the South China Sea: Insights into their different adaptabilities under contrasting environmental conditions. *Sci. China Earth Sci.* 64, 839–852 (2021).

Grover, R., Maguer, J. F., Reynaud-Vaganay, S. & Ferrier-Pagès, C. Uptake of ammonium by the scleractinian coral *Stylophora pistillata*: Effect of feeding, light, and ammonium concentrations. *Limnol. Oceanogr.* 47, (2002).

Muscatine, L., Porter, J. W. & Kaplan, I. R. Resource partitioning by reef corals as determined from stable isotope composition. *Mar. Biol.* 100, 185–193 (1989).

Tremblay, P., Grover, R., Maguer, J. F., Hoogenboom, M. & Ferrier-Pagès, C. Carbon translocation from symbiont to host depends on irradiance and food availability in the tropical coral *Stylophora pistillata*. *Coral Reefs* 33, 1–13 (2014).

Pupier, C. A. et al. Productivity and carbon fluxes depend on species and symbiont density in soft coral symbioses. *Sci. Rep.* 9, 17819 (2019).

COMMENT

Significant results include an increase in symbiont density between Site 1 (control) and Sites 2 and 3 (reduced pH). Chl a per unit area showed a similar pattern. These results suggest that production of photosynthetic assimilates would be greater in the corals found at Sites 2 and 3. The PAM data show no significant difference in photosynthetic efficiency of the symbionts across the gradient. This implies that the symbiont cells aren't acclimatizing to the reduced pH. Rather the coral host may be regulating the density of symbiont cells in such a way as to optimize the availability of photosynthetic assimilates under reduced pH conditions when energy expenditures for calcification and acid/base homeostasis will be higher. It is interesting that del ^{13}C data also support more translocation of photosynthetic assimilate in the Site 2 and 3 corals, although it is not clear to me that the difference (60% v 40%) is statistically significant. Are the authors able to test the statistical significance of this difference?

REPLY:

Agreed. In the most acidic Sites (Sites 2 and 3) we find distinct haplotypes compared to the control (Site 1). There is evidence in the scientific literature that increases in cell density can occur where densities are simply not optimal (i.e., at 'steady state' levels) and environmental conditions, such as relatively high pCO_2 (Sugget et al. 2012) and N_2 fixation (Marcelino et al., 2017; Rädicker et al., 2015; Santos et al., 2014), could favor growth (Jones & Yellowless 1997; Muscatine et al. 1998). Whether this acclimatory response is regulated by the coral and/or by the algal symbionts is not clear.

We ran an ANOVA (after checking for normality and homogeneity of variance) to test the difference in translocation of photosynthetic assimilate among Sites and it resulted statistically non-significant, likely due to the low N (N = 4 per site). Nonetheless, the raw data show the highest values in the lowest pH Site (Site 3), as shown in revised Figure 4 which now includes a box plot with these data. Changes to the text were made in Lines 389-390.

COMMENT

The massive reduction in host C:N ratio doesn't make much sense to me. If the C:N of the symbionts is increasing and there is an increase in the amount of that photosynthetic assimilate translocated to the host wouldn't you expect the C:N of the host to also increase (assuming that you are what you eat)? Also the extremely low C:N ratios of the host at Sites 2 and 3 seem questionable to me. Is it really possible for the bulk organic matter of coral host to have C:N composition of 1:1?

REPLY:

Partially agree. Under the field conditions of this study, we have a massive increase in N at the low pH sites driven by N_2 fixation in the coral tissue and mucus (Palladino et al. 2022) that flips the host tissue C/N trend in the opposite direction compared to the algal symbionts. Diazotroph dinitrogen fixation is known to considerably increase with acidification in open sea water (Wannicke et al., 2018), as well as in shallow coral reefs, where ocean acidification is associated with a general increase in the amount of nitrogen fixed (Cardini et al., 2014). The tissue C/N ratios found in *B. europaea* at the low pH Sites are unusually low compared to ratios found in our control Site, which match values generally found in marine organisms (from 6.5 to 8.7). However, C/N ratios can reach values around 2 in marine particulate organic matter (Xu et al. 2021; Crawford et al. 2015) and can even drop below 1 in sediments characterized by significant amounts of fixed nitrogen (Kikumoto et al. 2014). Thus, nitrogen fixation, mobilization, and storage found in *B. europaea* at the low pH sites (Palladino et al. 2022) provide a viable explanation for the massive increase in the denominator of the ratio. We thank the Reviewer for raising these points which stimulated us to enhance the discussion accordingly (Lines 493-515).

REVIEWERS' COMMENTS:

Reviewer #2 (Remarks to the Author):

The manuscript by Prada et al claims to show that the Mediterranean zooxanthellate scleractinian coral *Balanophyllia europaea* acclimatized to low pH/high pCO₂ conditions at a CO₂ vent off Panarea Island (Italy). This would be a significant finding because it is based on in situ observations of the coral living along a natural pH gradient.

The authors have done a good job of addressing my concerns in the earlier version of the manuscript. While the increase in C/N ratio of the symbionts in conjunction with increased N-supply due to increased N₂ fixation doesn't hang together comfortably in my mind the authors describe a scenario where both the host and the microbial N₂ fixers are competing for carbon assimilates from the symbionts which could lead to complicated interactions between the various processes. Overall I believe that the observations are interesting enough to recommend acceptance.